

# Synergistic effect of water-soluble species and relative humidity on morphological changes of aerosol particles in Beijing mega-city during severe pollution episodes

Xiaole PAN[1], Baozhu GE[1], Zhe WANG[1,2], Yu TIAN[1,3], Hang LIU[1,3], Lianfang WEI[1], Siyao YUE[1,3], Itsushi UNO[2], Hiroshi KOBAYASHI[4], Tomoaki NISHIZAWA[5], Atsushi SHIMIZU[5], Pingqing FU[1,3,7], Zifa WANG[1,3,6],

[1] State Key Laboratory of Atmospheric Boundary Layer Physics and Atmospheric Chemistry, Institute of Atmospheric Physics, Chinese Academy of Sciences, Beijing, 100029, China

[2] Research Institute for Applied Mechanics, Kyushu University, Kasuga, 816-8580, Japan

[3] University of Chinese Academy of Sciences, Beijing,100049, China

[4] University of Yamanashi, Yamanashi, 400–0016, Japan.

[5] National Institute for Environmental Studies, Ibaraki, 305-8506, Japan

[6] Center for Excellence in Regional Atmospheric Environment, Institute of Urban Environment, Chinese Academy of Sciences, Xiamen 361021, China

[7] Institute of Surface-Earth System Science, Tianjin University, Tianjin, 300072, China

*Correspondence to*: Xiaole PAN (panxiaole@mail.iap.ac.cn)

**Abstract.** Depolarization ratio (δ) of backscattered light from aerosol particle is an applicable parameter for real-time distinguishing spherical and non-spherical particles, which has been widely adopted by ground-based Lidar observation and satellite remote sensing. From November 2016 to February of 2017, it consecutively suffered from numbers of severe air pollution at Beijing with daily averaged mass concentration of $PM_{2.5}$ (aerodynamic diameter less than 2.5μm) larger than 150 μg/m$^3$. We preformed concurrent measurements of water-soluble chemical species and depolarization properties of aerosol particles on the basis of a continuous dichotomous Aerosol Chemical Speciation Analyzer (ACSA-14) and a bench-top optical particle counter with a polarization detection module (POPC). We found that δ value of ambient particles generally decrease as mass concentration of $PM_{2.5}$ increased at unfavorable meteorological condition. Ratio of mass concentration of nitrate ($NO_3^-$) to that of sulfate ($SO_4^{2-}$) in $PM_{2.5}$ was 1.5 ± 0.6, indicating of great importance of $NO_x$ in the formation of heavy pollution. Mass concentration of $NO_3^-$ in $PM_{2.5}$ (fNO3) was generally an order of magnitude higher than that in coarse mode



(cNO3) with a mean fNO3/cNO3 ratio of 14 ±10.  Relatively high allocation (fNO3/cNO3 = 5) of $NO_3^-$ in coarse mode could
be partially attributed to hygroscopic growth/coagulation of nitrate-rich fine mode particles under higher relative humidity
condition. As a result, δ values of particles with Dp = 2 μm ($\delta_{Dp=2}$) and 5 μm ($\delta_{Dp=5}$) decreased evidently as the mass fraction
of water-soluble species ($NO_3^-$ and $SO_4^{2-}$) increase in both $PM_{2.5}$ and $PM_{2.5-10}$, respectively. In particular, due to synergistic
effect of RH, $\delta_{Dp=5}$ value could decrease by 50% as mass fraction of $NO_3^-$ in $PM_{2.5-10}$ increased from 8% to 23%. It suggested
that alteration of non-sphericity of mineral dust particles was evident owing to coating with pollutants and heterogeneous
reactions on the surface of the particle during heavy pollution period. This study brings the attention to great variability of
morphological changes of aerosol particles along the transport, which have great complex effects in evaluating their climate
and health effect.

## 1 Introduction

Tropospheric aerosols, in particular the particles with aerodynamic diameter less than 2.5 μm ($PM_{2.5}$), have detrimental impact
on human health. It degrades the air quality by increasing atmospheric turbidity, and affects regional/global climate by
disturbing the solar radiation transfer in the Earth system via scattering/absorbing light directly and altering amount and
lifespan of cloud (Ramanathan et al., 2001; Kaufman et al., 2002). In the past decades, intensive anthropogenic/industrial
activities in East Asia emit huge amount of primary pollutants such as $SO_2$, $NO_x$, $NH_3$, VOCs etc., which resulted in severe
air pollution. $PM_{2.5}$ was mainly composed of light-scattering species (sulfate, nitrate, ammonium, organics etc.) and light-
absorbing matter (BC, BrC etc.). Under high relative humidity condition, most of aerosol particles absorb water vapor and
undergo apparently hygroscopic growth, which lead to dramatic changes in its mass concentration, size distribution, optical
properties (single scattering albedo etc.), as well as its corresponding morphologies. For example, freshly emitted rBC particles
were normally in a chain-like aggregates with visible carbon monomers, and they are sometimes fully coated or partially
coated by organics/inorganic matters. Due to atmospheric aging processes in polluted areas at high relative humidity rBC core
tend to shrink to a compact one with apparent non-refractory coatings, and its light-absorbing capacity for thickly coated rBC
also increased by a factor of 2.
Mineral aerosol is also one of the key compounds in East Asia, it was frequently reported to be coated by anthropogenic
pollutants along its transport owing to heterogeneous reactions with reactive acidic gas and coagulation of soluble particles.
In polluted area, photo-chemically formatted nitric acid ($HNO_3$) could easily react with $CaCO_3$ to form $Ca(NO_3)_2$ on the
surface of dust particles. The consecutive water-absorbing process will also lead to apparent morphological changes of dust
particles, which also impact on dust-cloud interaction (Tian et al., 2018) and new particle formation etc. Till now, online
investigation on the morphological changes of aerosol particle in the ambient environment is still limit. The widely-adopted
method to study the morphology and mixing state of particles is filter-based single particle sampling with electron-microscopy
inspection in laboratory. For example, scanning electron microscopy (SEM) coupled with energy dispersive X-ray
spectrometry (EDX) could provide information about surface topography, composition of the sample surface by scanning it
with a high-energy beam of electrons. Li et al., (2009) using Transmission electron microscopy (TEM) found that approximate





90% of sampled mineral particles were covered by visible nitrate coatings during pollution episode in Central East China. As a matter of fact, such hygroscopic coating on some individual dust particles was observed not only in polluted area but also at clean marine area. The observation on R/V during ACE-Asia found that dust particles mixed with chloride was sometimes dominant over sulphate and nitrate due to disassociation of acidified sea salt particles (Sullivan et al., 2007). Tobo et al., (2010) found that Asian dust particles could also be deliquescent to aqueous droplet as a result of formation of $CaCl_2$. Dust particle could also acquire sulphate coatings via either heterogeneous uptake of gaseous $SO_2$ and subsequent oxidation or coagulation within cloud or fog droplets (Kojima et al., 2006), as well as volatile carbonaceous species due to condensation processes (Kim and Park, 2012). Although diversity in chemical composition and structure of single particles with different degree of aging were investigated in the past studies (literatures), these analyses had to be inspected objectives one by ones, and such labor intensive operation cause difficult in broadening the results due to poor statistics (Li et al., 2009).

To obtain better understanding the real-time morphological variation of atmospheric processing particles, polarization property of backscattering light from the illuminated particle has been used as an applicable surrogate. For spherical particle, oscillation direction of magnetic wave of scattering light was identical to the incident light. Therefore, depolarization ratio (DR, here defined as the ratio of *s*-polarized to *p*-polarized backward component) was zero theoretically. However, for uncoated dust particles, the direction deviates significantly with a large DR value. Such characteristic is widely used to distinguish dust and spherical particles by both ground-based Lidar observation (Asian Dust and Aerosol Lidar Observation Network, AD-Net) (Müller et al., 2007; Thorsen et al., 2016; Hofer et al., 2017) and satellite on-board remote sensing (Cloud-Aerosol Lidar with Orthogonal Polarization on-board the CALIPSO) (Winker et al., 2010; Yu et al., 2015; Geng et al., 2011) presuming that spherical particles and dust were externally mixed. On the basis of this technique, studies on spatial-resolved distribution (Hara et al., 2008;Uno et al., 2008), transport pattern (Uno et al., 2009) of pollution and dust, and data processing algorithm (Nishizawa et al., 2007; Winker et al., 2009; Nishizawa et al., 2011) have been widely performed. For instance, Shimizu et al. (2004) summarized the contributions of different aerosol types on the total backscattering coefficient at multiple sites in East Asia using three-channel Mie scatting Lidar data. Huang et al. (2015) identified anthropogenic dust particles due to human activities and its contribution to global dust loading on the basis of CALIPSO observation. Recently, a multi-wavelength Mie-Raman Lidar (MMRL) and a new algorithm were developed to estimate extinction coefficients for black carbon, dust, sea salt, and air-pollution aerosols (a mixture of sulfate, nitrate and organic carbon substances)(Hara et al., 2017;Nishizawa et al., 2017).

It was worth noting that atmospheric particles were normally internally mixed. Once the dust particles were coated by other water-soluble pollutants, its polarization degree may decrease. Therefore, it is a challenge to accurately classifying in coated/uncoated dust particles due to its morphological diversity. In particular, Lidar adopted a volume depolarization ratio to discriminate different aerosol types; it was easily biased due to presence of small spherical particles in the volume of targeted air volume. To overcome this shortcoming, recently a bench-top optical particle counter equipped with a depolarization module (Polarization Optical Particle Counter) was developed to detect the size-resolved polarization of individual particles.



POPC is capable of investigating the temporal variation of mixing state of single dust particles. The observation at an urban site in Japan showed that DR of super-micron particles decreased evidently due to an increase of mass fraction of nitrate concentration during a long stagnant dust event (Pan et al., 2015). In Beijing, due to synergistic effect of water soluble substance at the surface of dust and relative humidity, the non-sphericity of dust particles even tent to be sphericalized (Pan et al., 2017;Wang et al., 2017). The long-term observation on such effect was still lacking.

From November 15, 2016 to Februarys 18, 2017, Beijing was consecutively suffered from several severe air pollutions with hourly averaged mass concentration of $PM_{2.5}$ and $PM_{2.5-10}$ larger than 300 $\mu g/m^3$ and 100 $\mu g/m^3$. It provided a good chance to investigate into the interaction between dust particles with pollutants. During this period, chemical composition, size distribution, polarization properties of aerosol particles were concurrently measured, as well as vertical profile of backscattering coefficient by Lidar at a downtown tower site of Beijing mega city. The objective of this study was to investigate depolarization properties of aerosol particles in the polluted urban site on the single-particle basis, and to study the impact of both water-soluble species and relative humidity (RH) on the morphological changes of dust particles. This study will provide useful information in better understanding the physical and optical properties of particles in East Asia, and improving numerical simulation on its environment/climate effect.

## 2 Observations

### 2.1 Observation overview

The field measurement was performed at a tower campus of Institute of Atmospheric Physics/Chinese Academy of Sciences in the downtown area of Beijing mega city. The observation site located between 3rd North Ring road and 4th North Ring road, where anthropogenic emissions are intensive in the daytime. Within the campus (150 m x 50 m), there is a 325 m tower for meteorological measurement and scientific research; therefore anthropogenic activity nearby is limited. During the observation period, a continuous dichotomous Aerosol Chemical Speciation Analyzer (ACSA-14) was placed on the roof of two-story building in the campus. Polarization Optical Particle Counter was placed in air-conditioned room on the roof to measure polarization properties of single particle less than 10 μm. To avoid loss of particles in coarse mode, the sampled air was drawn into the room by a supporting pump (flow rate: 10 lpm) through 2 m long 1/4-inch vertically assembled stainless steel tube. From November 15 to December 15 2016, a UK-China joint field campaign, entitle as "In-depth study of air pollution sources and processes within Beijing and its surrounding region (APHH-Beijing)" was also performed in the tower campus. Detailed information about the objectives and instruments are shown in webpage (https://www.atmos-chem-phys.net/special_issue932.html).

### 2.2 Instruments

In this study, mass concentrations of particulate matter and water-soluble chemical compounds in both $PM_{2.5}$ and $PM_{2.5-10}$ were measured by ACSA-14 (KIMOTO electric Co. Ltd, Osaka, Japan) with 1-hour intervals at the observation site. The mass concentration of particulate matter was determined using beta-ray absorption method. Mass concentration of $SO_4^{2-}$ was



determined on the basis of $BaSO_4$-based turbidimetric method with addition of $BaCl_2$ dissolved in polyvinylpyrrolidone
solution. Mass concentrations of $NO_3^-$ and water-soluble organic carbon were determined using ultraviolet absorption-
photometric method. Because mass concentration of $NO_3^-$ was general high in Beijing, the instrument generally collected
aerosol samples in the first 5~10 minute in each hour and analyzed the samples in the rest time to guarantee follow Beer-
Lambert Law. The acidity of particles [$H^+$], in unit of $nmol/m^3$, was semi-quantitatively determined using pH-indicator
absorption-photometric method. The basic equation is $pH_{solution} = -\log [H^+ \times 10^{-6} + 10^{-4.6}]$, presuming that all the water-soluble
matter was dissolved in the extract liquid with pH value of 4.6. A factor of $10^{-6}$ was used to convert unit of [$H^+$] from $nmol/m^3$
to mol/L. The comparison of [$H^+$] in $PM_{2.5}$ between ACSA and off-line filter-pack measurement showed a good linear
correlation ($[H^+]_{ACSA} = 3.33 + 0.81 \times [H^+]_{FP}$) with $r^2 = 0.54$ (Personal communication with Prof. Osada in Nagoya University).
The details of the ACSA instrument are described in literature (Kimoto et al., 2013).
Depolarization ratio of single particle was determined using a Polarization Optical Particle Counter (POPC). POPC adopted a
780 nm linearly polarized laser beam to illuminate the aerosol particles that passed through measuring chamber vertically.
The direction of vibration of the electric field of the incident laser is parallel to the plane of the scattering angle. Detailed
information about POPC was described in literatures (Kobayashi et al., 2014;Pan et al., 2016;Pan et al., 2017). Forward
scattering signal at 60 degree respect to direction of incident light was measured by a photodiode with acceptance angle 45
degree to determine the size of particle. The backward scattering signal at 120 degree was split into P component (horizontal
with respect to the plane of the scattering angle) and S component. The depolarization ratio (DR) of the particles was defined
as the ratio of S component to P component (S/P). To avoid the coincidence error of the measurements, the inlet flow rate of
POPC was set to 80 ccm and was diluted with zero air (920 ccm, RH = 38 ± 1%). Overall measurement uncertainty in size
determination was less than 15%.
During observation period, the vertical structure of backscattering coefficient for aerosols was derived from Mie-scattering
Lidar system that installed at the same place of ACSA-14. This Lidar system, developed by the research group in National
Institute for Environmental Studies (NIES), employs a flash-lamp-pumped Q-switching Nd:YAG laser as the light source. It
emits pulsed lights with wavelength of 1064 nm and 532 nm with at a frequency of 10 Hz, and collects the backscattered light
from the atmosphere by a 20 cm Schmidt-Cassegrain telescope. The light at 532-nm wavelength is also further separated into
S and P component (Sugimoto et al., 2002). The algorithm for classifying sphere and dust particles was described in literatures
(Shimizu et al., 2004; Nishizawa et al., 2011;Shimizu et al., 2017). To be note that, direct comparison of depolarization ratio
between POPC and Lidar system is difficult. Firstly, Lidar system receives backscattering light at almost 180 degree with a
field of view of 1 mrad; whereas POPC employs 120-degree backscattering signal. Secondly, Lidar system measures the total
volume depolarization from a volume of targeted air parcel; whereas depolarization properties from POPC is on a single
particles basis.



**2.3 Footprint analysis**

We simulated footprint region of aerosol particles at the observation site using NOAA Hybrid Single Particle Lagrangian Integrated Trajectory (HYSPLIT) model (v4.9; available at http://ready.arl.noaa. gov/HYSPLIT.php). This model has been widely applied into calculating long-range or meso-scales transport and footprint regions of air pollutions with fast computational speed and high spatial resolution. HYPLIT model is capable of forward/backward run in time to simulate the dispersion/potential source of tracers in a given location. Detailed description and validations of this model was seen in literature. In this study, input meteorological data of model is the product (GDAS dataset) of Global Forecast System (GFS) from National Centers for Environmental Prediction (NCEP) with a spatial resolution of 0.5 by 0.5 degree, and a time-resolution of 3 hours (0000, 0600, 1200 and 1800 UTC from data assimilation product; 0300, 0900, 1500 and 2100 UTC from forecast model). By offsetting the release point by a meteorological grid point in the horizontal and 0.01 sigma units in the vertical, ensemble simulation have 27 trajectories were calculated simultaneously in each hour, providing great advantage in evaluate the uncertainty and potential footprint region. In the present study, the grids at which height of backward endpoint of air parcel was less than the height of mixing layer was labeled as potential footprint region.

**3 Results and discussions**

**3.1 Overview in particulate matters and chemical species**

**3.1.1 Comparison of mass concentration of particulate matters**

Figure 2a shows the temporal variations of mass concentrations of ambient $PM_{2.5}$ that were measured by ACSA-14 at the observation site. For comparison, mass concentration of $PM_{2.5}$ observed at a state-controlled monitoring station (Olympic Centre, about 3 km northeast of the LAPC site) was plotted in the figure. In general, two results were in consistent well with $r^2 = 0.8$, indicating that the pollution events were generally in regional scale with minor interference from emission sources nearby. During observation period, number size distribution of ambient particles with optical diameter between 0.3 μm - 10 μm were measured with POPC, and the mass concentration of $PM_{2.5}$ was estimated assuming that all the particles were spherical with a density of 1.77 g/cm$^3$. Mass concentration of $PM_{2.5}$ estimated by POPC was compared well with ACSA-14 measurement before January 7, 2017, when ambient relative humidity (RH) was almost above 40%. Whereas estimated mass concentration of $PM_{2.5}$ was underestimated obviously. One possible explanation is that, both air temperature and ambient RH after January 7, 2017 decreased evidently. The interaction between water-soluble matters in $PM_{2.5}$ and moisture were unlikely occurs, and the hypothesis of spherical shape of particles due to hygroscopic may resulted in relatively larger uncertainty in estimating the optical size of particle on the basis of scattering light. Another possibility is that the chemical composition of particles are different (discussed in section 3.1.2). As shown in Figure 2d, the northerly wind become stronger, which resulted in larger proportion of mineral dust matters, the latter which have larger density (2.2 - 2.8 g/cm$^3$). Deploying the same density for both particle in fine mode and coarse mode will lead to underestimation of total mass. It was pronounced for the case in January 29, 2017 that observation site was subject to floating dust event with a hourly averaged mass concentration of $PM_{10}$



reached 734 µg/m$^3$. Daily averaged mass concentration of PM$_{2.5}$ measured by ACSA-14 was 354.3 µg/m$^3$, four times higher
than the values (77.1 µg/cm$^3$) estimated by POPC. Deployment of a larger particle's density of 2.8 g/cm$^3$ only explain 35% of
the difference. Low detection limit of particle size of POPC was 0.5 µm, and miscounting of particles less than 0.5 µm was
estimated to contribute another 10% of the difference. The most possible reason is that the irregularity of particles in fine
mode resulted in significant underestimating in particle diameter on the basis of scattering signal. However, the estimation of
spherical particles diameter by POPC bears smaller bias theoretically. For example, during a typical anthropogenic pollutant
dominant case on January 1, 2017, mass concentrations of PM$_{2.5}$ and PM$_{10}$ was as high as 438.8 and 626.2 µg/m$^3$, respectively.
Hourly averaged mass concentration of PM$_{2.5}$ estimated by POPC correlated well with ACSA-14 measurement with a ratio of
1.1 ± 0.1. It was noticeable that PM$_{2.5}$ accounted for 80% of PM$_{10}$, and ambient relative humidity was 65%. It was implied
that anthropogenic water-soluble compounds in PM$_{2.5}$ underwent hygroscopic processes that may alter asphericity of the
particles (detail discussion in section 3.5).
Regarding the particles in coarse mode, mass concentration estimated by POPC is much better than that in fine mode. It was
because, firstly, the detection efficiency of POPC for the particles in coarse mode was better than that in fine mode, and
miscounting of particles in coarse mode by POPC was less likely to occur; secondly, interference of asphericity of particles
in coarse mode was insignificant in determining size of particle according to scattering signal. Besides, change in the refractive
index of particles due to mixing of mineral dust particles with anthropogenic pollutants (such as black carbon) also have
limited impact in size determination. In general, daily averaged mass fraction of PM$_{2.5-10}$ in PM$_{10}$ (PM$_{2.5-10}$/PM$_{10}$) was ranging
0.25 ~ 0.7 (Supplementary Figure). The minimum value occurred in severe pollution days when daily averaged mass
concentration of PM$_{2.5}$ was larger than 250 µg/cm$^3$ (Air Quality Level: VI), whereas PM$_{2.5-10}$/PM$_{10}$ ratio increased as mass
concentration of PM$_{2.5}$ decrease. It was because formation of secondary particulate matters (such as sulfate and nitrate) during
pollution episode was so overwhelming that make the contribution of mineral dust decrease, although mass concentration of
PM$_{2.5-10}$ increased. Number of studies have addressed the importance of mineral dust in promoting new particle formation (Nie
et al., 2012) and conversion of SO$_2$ to sulfate (He et al., 2012), both of which was related to formation of OH radical. However,
this study was performed in winter, role of mineral dust in the formation of regional pollution is out of scope of this study.
**3.1.2 Chemical compounds in fine and coarse mode**

In winter, nitrate, sulfate and water-soluble organic carbon (WSOC) were found in both fine mode and coarse mode (Figure
3). Mass concentration of nitrate in the fine (fNO3) was 28.3 ± 33.7 µg/m$^3$ averagely with a maximum value of 190.9 µg/m$^3$.
Mass concentration of nitrate in the coarse mode (cNO3) was 2.9 ± 4.8 µg/m$^3$. The maximum value of cNO3 (43.1 µg/m$^3$)
occurred at the different time from that for fNO3, implying of complicated mass equilibrium of nitrate among different size
range. Table 1 summarizes the fNO3/cNO3 ratio as a function of mass concentration of PM$_{2.5}$ at RH > 40% and RH < 40%.
We found that, when ambient RH was less than 40%, fNO3/cNO3 ratio has a positive correlation with mass concentration of
PM$_{2.5}$, which demonstrated that nitrate mass preferentially formed in fine mode and mass transfer of nitrate toward coarse





mode was unlikely happen due to weak hygroscopicity of particles; However, when ambient RH was larger than 40%, fNO3/cNO3 ratio did not increase with $PM_{2.5}$ concentration with a mean of $12.4 \pm 6.5$. The possible explanations were, first, once the aerosol phase nitrate formed in the fine mode, it absorbed water vapor simultaneously and grew larger; secondary, ambient nitric acid may directly stick on the surface of particles in coarse mode through heterogeneous processes; Although high concentration of $PM_{2.5}$ in Beijing was regard as synergistic contributions from both local formation and long-range transport, it did not influence the fNO3/cNO3 ratio, at least in the present study.

Mass concentration of sulfate in the fine mode (fSO4) and in the coarse mode (cSO4) were $18.9 \pm 24.8$ $\mu g/m^3$ and $2.2 \pm 2.5$ $\mu g/m^3$ with maximum values of $143.1$ $\mu g/m^3$ and $25.2$ $\mu g/m^3$, respectively. As shown in Figure 3b, the variabilities of fSO4 and cSO4 concentrations have the same trend. fSO4/cSO4 ratio increased with mass concentration of $PM_{2.5}$, irrespective of ambient RH, because sulfate in the fine mode mainly formed via homogeneous reaction between sulfuric acid with ammonia. The positive correlation between fSO4/cSO4 ratio and RH indicated that water vapor also took effect in the formation of sulfate. It was worthy to note that, fNO3/fSO4 ratio varied among cases due to different air mass origin and meteorology condition. For instance, on Dec 21, 2016 (blue shaded strip in Figure 3), mass concentration of fNO3 and fSO4 were $122.4$ $\mu g/m^3$ and $74.1$ $\mu g/m^3$, with a fNO3/fSO4 ratio of 1.6; however on Jan 1, 2017 (red shaded strip in Figure 3), mass concentration of fSO4 increased to $125.2$ $\mu g/m^3$ and fNO3 decrease to $99.2$ $\mu g/m^3$ with a fNO3/fSO4 ratio of 0.79. Backward trajectory analysis for the case on Jan 1, 2017 showed that the air mass was mainly transport from North China Plain along Taihang Maintain, coal burning in NCP in winter may contribute to the sharp increase of fSO4; however, for the case on Dec 21, 2016, the air mass was mostly stagnant near Beijing area, and the intensive emission of NOx resulted in the large difference between fNO3 and fSO4. The variability of WSOC in both fine mode (fWSOC) and coarse mode (cWSOC) were consistent with fSO4 with maximum values of $167.6$ $\mu g/m^3$ and $8.7$ $\mu g/m^3$ (Figure 3c).

Mass fraction of total water soluble matters (WSM) in $PM_{2.5}$ and acidity of the particle are shown in Figure 3d. Here, WSM includes only $SO_4^{2-}$, $NO_3^-$, WSOC, and $NH_4^+$, the latter of which was estimated on the basis of equation ($[NH_4^+]$ = $18 \times ([SO_4^{2-}]/96 \times 2 + [NO_3^-]/62 \times 2 - [H^+]/1000)$. In general, mass fraction of WSM in $PM_{2.5}$ during pollution period was higher than that during clean period. On average, WSM/$PM_{2.5}$ was $0.5 \pm 0.16$. We found that, WSM/$PM_{2.5}$ was more likely related to origin and residence time of air mass that the ambient loading of $PM_{2.5}$ concentration. For example, daily averaged mass concentration of $PM_{2.5}$ reached the maximum ($447.5$ $\mu g/m^3$) on Jan 1, 2017; whereas maximum value of WSM/$PM_{2.5}$ ($0.80$) occurred on Jan 7, 2017 when the pollution period almost ended. On Jan 1, 2017 observation site was prevailing southwesterly wind which introduced pollutants from NCP where industrial emission was intensive; However, air mass was mostly from east region on Jan 7, 2017, and high concentration of cNO3 (Figure 3a) and high RH (Figure 2d) indicated that heterogeneous processes played a key role. Mole concentration of $[H^+]$ in fine mode (fH+) correlate well with mass concentration of fSO4 (Figure 3d), implying of possibility of presence of surplus of sulfuric acid that converted from $SO_2$ that emitted from coal burning during heating period.




Table 1 The relationship between fNO3/cNO3, fSO4/cSO4 and mass concentration of PM$_{2.5}$ at different RH condition.

| RH Classification | PM$_{2.5}$ Classification | PM$_{2.5}$ (µg/m$^3$) | | fNO3/cNO3 (unitless) | | fSO4/cSO4 (unitless) | | RH (%) | |
|---|---|---|---|---|---|---|---|---|---|
| | | Avg. | S.D. | Avg. | S.D. | Avg. | S.D. | Avg. | S.D. |
| RH<40% | <25 | 16.3 | 6.3 | 12.5 | 6.8 | 3.6 | 2.2 | 29.3 | 4.7 |
| | 25-50 | 37.5 | 7.2 | 15.1 | 7.2 | 4.7 | 1.9 | 32.3 | 5.9 |
| | 50-75 | 59.9 | 7.8 | 17.5 | 8.5 | 5.6 | 1.8 | 33.4 | 6.8 |
| | >75 | 98.2 | 23.4 | 26.2 | 12.1 | 6.2 | 1.8 | 32.2 | 6.4 |
| RH>40% | <50 | 33.4 | 13.1 | 12.1 | 6.1 | 4.5 | 1.8 | 53.3 | 14.1 |
| | 50-100 | 82.1 | 8.3 | 13.0 | 9.6 | 5.6 | 2.3 | 54.4 | 10.6 |
| | 100-150 | 119.2 | 14.5 | 9.5 | 5.3 | 6.1 | 3.4 | 60.2 | 10.1 |
| | 150-200 | 175.6 | 14.5 | 12.9 | 5.9 | 7.8 | 2.7 | 59.8 | 6.8 |
| | >200 | 290.9 | 61.6 | 14.3 | 5.3 | 10.3 | 2.0 | 61.1 | 9.2 |


### 3.2 Volume size distribution and depolarization ratio (δ)

Volume size distribution of aerosols and averaged depolarization ratio (δ) during observation period are shown in Figure 4. In
general, evolution of all the pollution cases were well captured by POPC. Volume size distribution of aerosols showed a
pronounced peak at 1 µm, at which secondary anthropogenic pollutants were dominant. Correspondingly, δ value of particles
were normally less than 0.1, consistent with the previous study in East Asia (Pan et al., 2016). On Nov 19, 2016, two peaks
were observed at size range of 1 µm and 2~3 µm. δ value of particles at two size were as low as 0.1. Such pattern of volume
size distribution has been described in previous study (Pan et al., 2015, 2016), which classified such phenomena as a mixing
pollution type that both anthropogenic pollutants and mineral dust particles interacted. Five-day trajectory analysis indicated
that air mass mainly originated from Mongolia Plateau and they were stagnant over East China for days (supplementary
figure). Volume peak at size of 4~5 µm was not observed probably due to fast gravity settling of large particles in coarse mode
during their stagnancy. On Nov 26, 2016, the observation site was influenced by floating dust, and volume size distribution
had a pronounced peak at 4 ~ 5 µm, as expected. δ value of submicron particles also increased to 0.3, implying of the substantial
presence of irregular mineral particles in the fine mode. From Dec 30, 2016 to Jan 8, 2017, it had a long-lasting pollution
period in Beijing. Volume size distribution of aerosols varied due to impacts from both change in Planet Boundary Layer
height and origin of aerosols. Multi-peak fitting analysis indicated that volume size distribution had two peaks with a dominant
peak at size of 0.9 µm and another peak at size of 2 µm. Different from case on Nov 19, 2016, the δ values of particles were
0.2~0.4, though ambient RH was the same (> 60%). It implied that chemo-physical properties of the particles were different,
probably due to impact by mineral dust. Detailed discussion was in section 3.3.2.

### 3.3 Variability of δ value for particles of different size
#### 3.3.1 δ value for particles in fine mode

As pointed out in previous study, δ value generally depends on the size of particle (SF2). Histogram analysis on the particles
at size of 1 µm showed that there was one peak mode at a $\delta_{Dp=1}$ value of 0.11; For the particles at size of 2 µm, a multi-



Gaussian fitting for frequency distribution of $\delta_{Dp=2}$ value showed there were one dominant peak at a value of 0.17 and a shoulder value of 0.23, the latter of which was mostly related to the impact of dust event. We found that variability of $\delta_{Dp=2}$ value of particles in fine mode was a synergistic effect of both water-soluble inorganic matter (WSI) and RH. For example, in Figure 5, $\delta_{Dp=2}$ value of particles decreased gradually from 0.3 to 0.1 as mass fraction of WSI in $PM_{2.5}$ increase from 0.2 to 0.65, and RH increased from 38% to 85%. It notes that, $\delta_{Dp=2}$ value decreased only when RH > 60% and mass fraction of WSI > 0.6. Mass fractions of both fSO4 and fNO3 in $PM_{2.5}$ showed a negative correlation with $\delta_{Dp=2}$ value with a slope of – 0.3 and -0.1, respectively. We speculated that fNO3 might play a key role in decreasing of $\delta_{Dp=2}$ value of particles in the fine mode because fNO3 was accounting for ~50% of total WSI and deliquescent point of ammonium nitrate was ~60%, and the impact of ammonium sulfate was less important since it started to undergo hygroscopic growth only at higher RH (79%). Quantitatively distinguishing respective contribution of fNO3 and fSO4 on the decrease of δ value of particles was difficult in the present study.

### 3.3.2 δ value of mineral dust aerosols

For the particles at size of 5 μm, histogram analysis of $\delta_{Dp=5}$ value had a wide range 0.3 - 0.55. Laboratory experiments on typical spherical particles at a size of 5.124 μm (SS-053-P) showed that their $\delta_{Dp=5}$ value was 0.07 ± 0.01. The larger $\delta_{Dp=5}$ value (> 0.3) of mineral dust particle in the ambient air indicated that they were aspherical in shape. Figure 6 showed time variation of vertical profile of extinction coefficient of dust particles derived from ground-based Lidar observation. We can see that there was a typical dust event on Dec 26, 2016 with a $PM_{2.5-10}/PM_{10}$ ratio of 68%, and extinction coefficient of dust particles at site was larger than 0.3 $Km^{-1}$. $\delta_{Dp=5}$ value of dust particle was varying around 0.5, and no decrease of δ value of dust particles was observed due to low cNO3 concentration and low RH (Figure 6). Another dust impact case was from Jan 1, 2017 to Jan 7, 2017, we found that $\delta_{Dp=5}$ value of dust particles were apparently low with a mean value of 0.35. In particular, during the period that mass concentration of cNO3 increased up to 10 μg/m$^3$ and ambient RH was ranging above 60%, hourly averaged $\delta_{Dp=5}$ value of dust particles decreased to 0.2, implying that cNO3 on the surface of dust particle may form as $Ca(NO_3)_2$ owing to heterogeneous processes, and consecutive hygroscopic growth resulted in the decrease of its $\delta_{Dp=5}$ value. Compared with the case on Dec 26, 2016, impact of this dust event was relative weak with a $PM_{2.5-10}/PM_{10}$ ratio of 27%. It was worth noting that all the mineral dust impacting cases were captured by both Lidar and POPC observations, nevertheless, the mixing state of mineral dust particles could be illustrated better with POPC measurement according to their $\delta_{Dp=5}$ variations. Once the morphology of mineral dust particles was modified due to cNO3 coating at high RH condition, Lidar observation may underestimate the impact of dust due to decrease of $\delta_{Dp=5}$ value of particles. Such phenomena maybe more likely to happen at the downstream of polluted area. For instance, dust particles were found to be spherical due to interaction with HCl and $HNO_3$ at marine area (Tobo et al., 2012). Observations using POPC at Kyushu area of Japan also indicated that there was a large amount of larger particles with a $\delta_{Dp=5}$ value between 0.05 ~ 0.15 when air mass came from NCP of China, implying that the morphology of dust particles were altering with transport (Pan et al., 2015, 2016).






### 3.4 Footprint analysis of mineral dust particles


As discussed above, the decrease of $\delta_{Dp=5}$ value of mineral dust particles were influenced by both mass concentration of cNO3
and ambient RH. We choose two mineral dust-influencing episodes to demonstrate the impact (Figure 7). Here, footprint
region of the air mass was calculated for the period that hourly averaged mass concentration of cNO3 was higher than 5 μg/m$^3$.
For the episode between Nov 24 and Dec 7 2016, the footprint region of air mass covered west of Inner Mongolia province
and South of Hebei province. The particles in coarse mode seemed to mostly related with anthropogenic dust (defined as dust
aerosols due to human activity, such as agriculture, industrial activity, transportation etc.) in NCP where NOx emission and
atmospheric nitrate loading were also high. Averaged RH was relatively low with a mean value of 30% (Figure 7c). However,
for the episode between Jan 1 and Jan 8, 2017, averaged RH within the footprint region around NCP was as high as 60%
(Figure 7d), although mineral dust particle had similar origins. Adsorption of water vapour and consecutive heterogeneous
reaction lead to obvious decrease of $\delta_{Dp=5}$ value of mineral dust particles. It indicated that a synergetic effect of both nitrate in
coarse mode and high RH condition lead to morphological changes in dust shape. The variability of morphological changes
was simulated on the basis of T-matrix methodology and randomly oriented elongated ellipsoid particles. We found that
observed maximal $\delta_{Dp=5}$ value (0.5) of dust particle corresponded to an aspect ratio (defined as ratio of the longest dimension
to its orthogonal width) of 1.7. When $\delta$ value of mineral dust particles decrease to 0.2, the aspect ratio was estimated to be 1.5,
not being "spherical". Therefore, we considered such dust particles as being 'quasi-spherical'. Huang et al., (2015) pointed
out that layer-integrated $\delta$ value of anthropogenic dust particles in the PBL of NCP was lower than that of Taklimakan dust
on the basis of CALIPSO Lidar measurement, mostly due to mixed with other more spherical aerosol within the PBL. Because
Lidar observation just provide a averaged $\delta$ value of all the particle in the detecting volume, external mixing of dust particle
with substantial amount of secondary anthropogenic particles in spherical shape could also result in a low $\delta$ value. From view
point of this study, irregularity of anthropogenic dust particles in NCP were possibly the same as the nature dust in the source
region, however, their interaction in particular at high RH condition will obviously lead to decrease in $\delta$ value.

### 3.5 Impact of heterogeneous reaction on $\delta$ value of particles


As shown in Figure 8, $\delta$ value of particles decrease obviously with increase of mass fraction of water soluble Matters (WSM),
in particular at high RH condition. For the particles at Dp = 2 μm, their $\delta_{Dp=2}$ value was normally 0.3 when mass fraction of
WSM in PM$_{2.5}$ was less than 0.2, whereas it decreased to 0.1 when the mass fraction of WSM increased to 0.6, and RH also
increased to 80% (Figure 8a). The linear relation was because the growth of particles in fine mode was closely related to
formation of secondary inorganic matters (sulfate, nitrate, etc.) and organic aerosols. Throughout the atmospheric chemistry
processes, the positive feedback between aerosol and water vapor was ubiquitous. For example, hygroscopic processes of



anthropogenic secondary inorganic resulted in abundance of aerosol liquid water content, and the latter of which could provide
an efficient media of multiphase reaction which promote new particle formation, and so on. Therefore, the particles in $PM_{2.5}$
generally approach to spherical in shape, resulted in a low $\delta_{Dp=2}$ value, and the high RH and the high possibility that spherical
particles formed. The morphological changes of particles in $PM_{2.5}$ could be well simulated on the basis of T-matrix method
(Pan et al., 2017).
For mineral dust particles, As discussed, the $\delta_{Dp=5}$ value (0.3~0.5) was clearly higher than that of particle in fine mode because
of this high irregularity (Figure 8b). Since the mass concentration of cSO4 and cNH4 were insignificant in coarse mode, the
$\delta_{Dp=5}$ value was plotted versus mass fraction of cNO3 in $PM_{2.5-10}$. Decrease of $\delta$ value of dust particles as a function of mass
fraction of cNO3 in $PM_{2.5-10}$ was also obvious, especially when RH was high than 60%. Numbers of studies have reported
emission of anthropogenic dust in NCP was significant, and Calcium was the most abundant crustal element in NCP. We
believed that that $Ca(NO_3)_2$ present on the surface of mineral dust particles, and the decrease of $\delta_{Dp=5}$ value of particle in coarse
mode was mostly due to heterogeneous reactions and consecutive water-absorbing processes. Such kind of mineral dust
particles coated with anthropogenic pollutants have been observed by numbers of previous electro-microscopic studies
(literatures). Previous studies also demonstrated the presence of cSO4 on the surface of dust particles, nevertheless, we think
the effect of cSO4 on the decrease of $\delta_{Dp=5}$ value was negligible because the mass fraction of cSO4 was tiny (<0.02) and the
$CaSO_4$ was hard to dissolve in limited amount of aerosol liquid water content. To note that, the chemical compounds (such
as kaolinite, illite, humic matters, etc.) of particles in coarse mode was very complicated, $\delta_{Dp=5}$ value just indicate a
synthetically effect of morphological changes as a result of all physical and chemical processes. To quantitatively characterize
their optical and environmental effect, detailed studies on $\delta_{Dp=5}$ value variability of one-fold compound in the laboratory was
still essentially need.

## 4 Conclusions and implications

Depolarization properties of aerosol particles is an important parameter classifying the aerosol types and describing the
variability of morphology of particles, which was related to complicated mixing processes and heterogeneous reactions. It
also has great impact on its transport and regional/global climate due to alteration of optical properties of particles. In 2017, a
field joint campaign (In-depth study of air pollution sources and processes within Beijing and its surrounding region, APHH-
Beijing) was performed at an urban site in Beijing mega city. One of key aims of the project is to assess the processes by
which pollutants are transformed through atmospheric chemical reactions. Taking this opportunity, we performed an online
observation on the depolarization ratio of single particles using a Polarization Optical Particle Counter. The chemical
compositions ($SO_4^{2-}$, $NO_3^-$, WSOC) and acidity of particles in both fine mode ($PM_{2.5}$) and coarse mode ($PM_{2.5-10}$) was
determined using a continuous dichotomous Aerosol Chemical Speciation Analyzer (ACSA-14). The main conclusions are as
follows: (1) Depolarization ratio ($\delta$) of ambient particles generally increase with its size due to increase in irregularity of the
particles, and the characteristic values of $\delta$ for the particles at Dp = 1 μm and 2 μm were 0.11, 017, respectively. Once the





observation site was influenced by dust event, both of δ values increased to above 0.3 due to presence of submicron mineral
dust particle in irregular shape. The δ value of the particles at Dp = 2 μm was mainly determined by the mass fraction of water-
soluble inorganic matter in PM$_{2.5}$, in which water vapor was fully involved in their atmospheric formation processes. (2) In
NCP, anthropogenic dust was an important contributor to the atmospheric loading of particles, their δ values (0.2-0.3) were
found to be smaller than that (0.5) of nature dust because of adsorption of acidic substance (such as HNO$_3$) and coagulation
with water-soluble anthropogenic pollutants (nitrate, sulfate) and consecutive chemical reactions on the surface of dust.
Ambient relative humidity plays a key role in altering the morphology of mineral dust as a result of hygroscopic processes of
deliquescent substances such as Ca(NO$_3$)$_2$, in particular when RH > 80%. In this study, we found that δ values of mineral dust
particle in NCP could be as smaller as 0.2, which could be termed as "quasi-spherical". (3) We found that allocation of
anthropogenic pollutants in fine and coarse mode was influenced by the RH along the trajectories of air mass, and increase of
nitrate mass in coarse mode was highly associated with the dust event. It indicated that the mineral dust particles in NCP was
mostly coated with anthropogenic pollution, upon which classification and quantitative determination of anthropogenic dust
emission was possible, though a pioneering study have been done on the basis of satellite remote sensing and land type (Huang
et al., 2015).
In this study, we provided solid evidences in morphological changes of mineral dust particles in NCP. Variability of δ value
of particles is also a valuable parameter in distinguishing how mineral dust particle interacted with anthropogenic pollutants
in formation of regional-scale haze pollutions. This result also spurs us to revisit decades of Lidar observation data in better
describing the transport and vertical distribution of Asian dust and pollution, and its regional environmental effects under the
scenarios of China thirty years' rapid urbanizations. A reliable optical model capable of discriminating multiple aerosol types
was necessary for detailed analysis of polarization-related remote sensing observations. This study suggested that an integrated
observation network with single-particle-based depolarization measurement was necessary for synthetically understanding
chemo-physical properties of Asian dust issue.
**Acknowledgements**
This work was supported by the National Natural Science Foundation of China (Grant No. 41675128 and 41571130034),
and in part supported by Chinese Ministry of Science and Technology (2016YFC0207904, 2017YFC0212402) and CAS
Information Technology Program (Grant No. XXH13506-302). The authors gratefully thank Prof. Kobayashi from
Yamanashi University and Prof. Yele Sun from State Key Laboratory of Atmospheric Boundary Layer Physics and
Atmospheric Chemistry/Institute of Atmospheric Physics for their valuable comments on the original manuscript.

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



# Figures



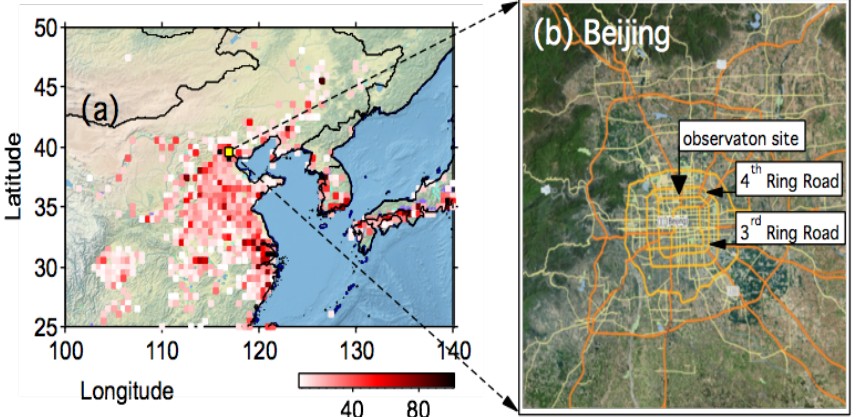


Figure 1. Geographic location of observation site and NOx emission in East Asia (a), the transportation map of Beijing mega
city and location of tower campus of IAP.




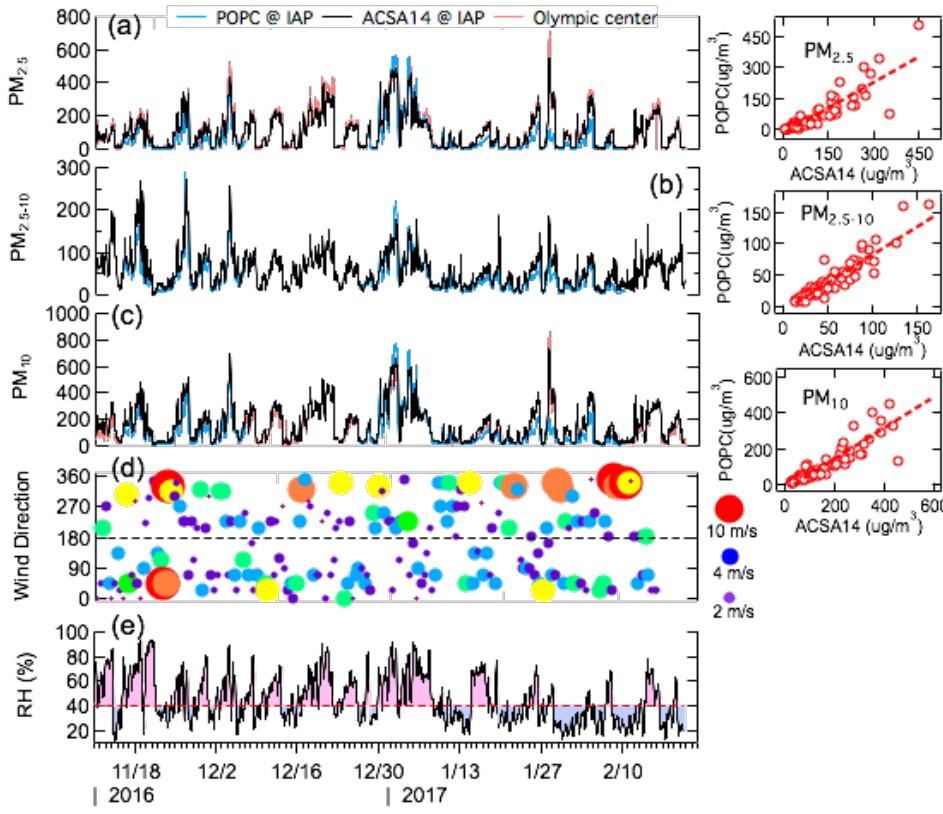

Figure 2.  Temporal variation of mass concentration of $PM_{2.5}$ (a), $PM_{2.5-10}$ (b), $PM_{10}$ (c) measured by ACSA14 and derived
from POPC measurement, wind speed and direction (d) and relative humidity (e) at observation site. The three scatter plots in
the right indicate the linear relationship between ACSA14 and POPC, the correlations are $Y_{POPC} = -6.8 + 0.89 \times X_{ACSA14}$ ($r^2 = $
0.86), $Y_{POPC} = -19.5 + 0.84 \times X_{ACSA14}$ ($r^2 = 0.77$), $Y_{POPC} = -27.7 + 0.87 \times X_{ACSA14}$ ($r^2 = 0.81$) for $PM_{2.5}$, $PM_{2.5-10}$ and $PM_{10}$
respectively.



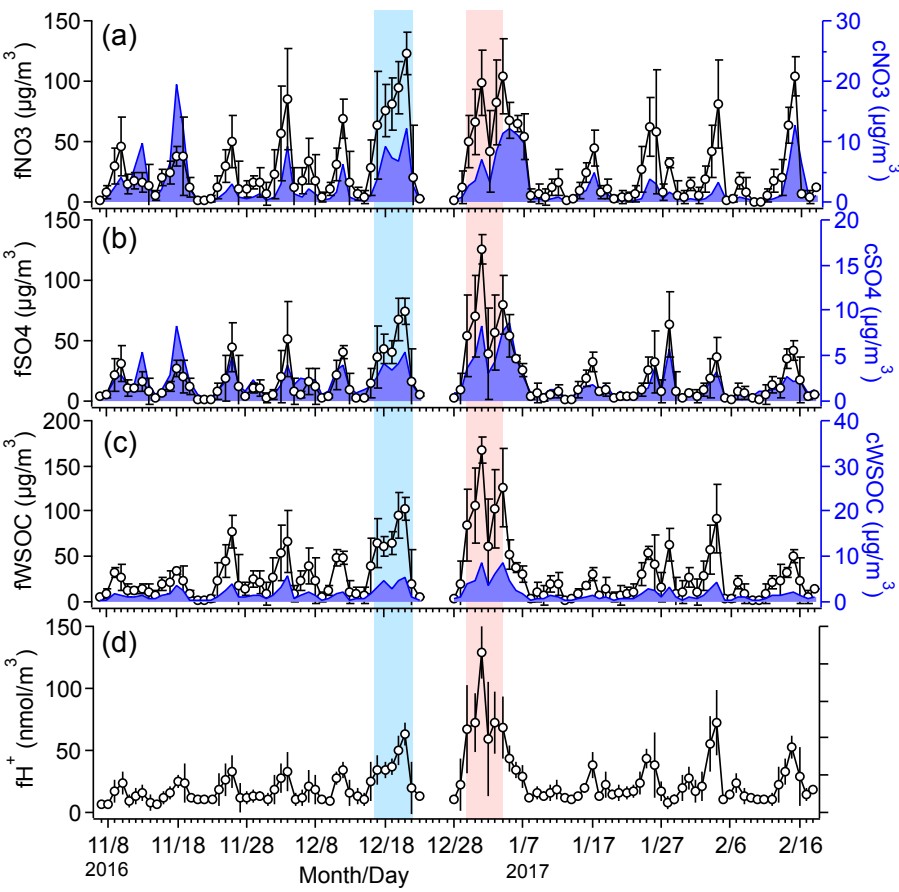

Figure 3 Temporal variations of mass concentrations of nitrate (a), sulfate (b) and water-soluble organic matters (c) in fine mode and coarse mode, and acidity of particle (fH$^+$) in the fine mode (d).



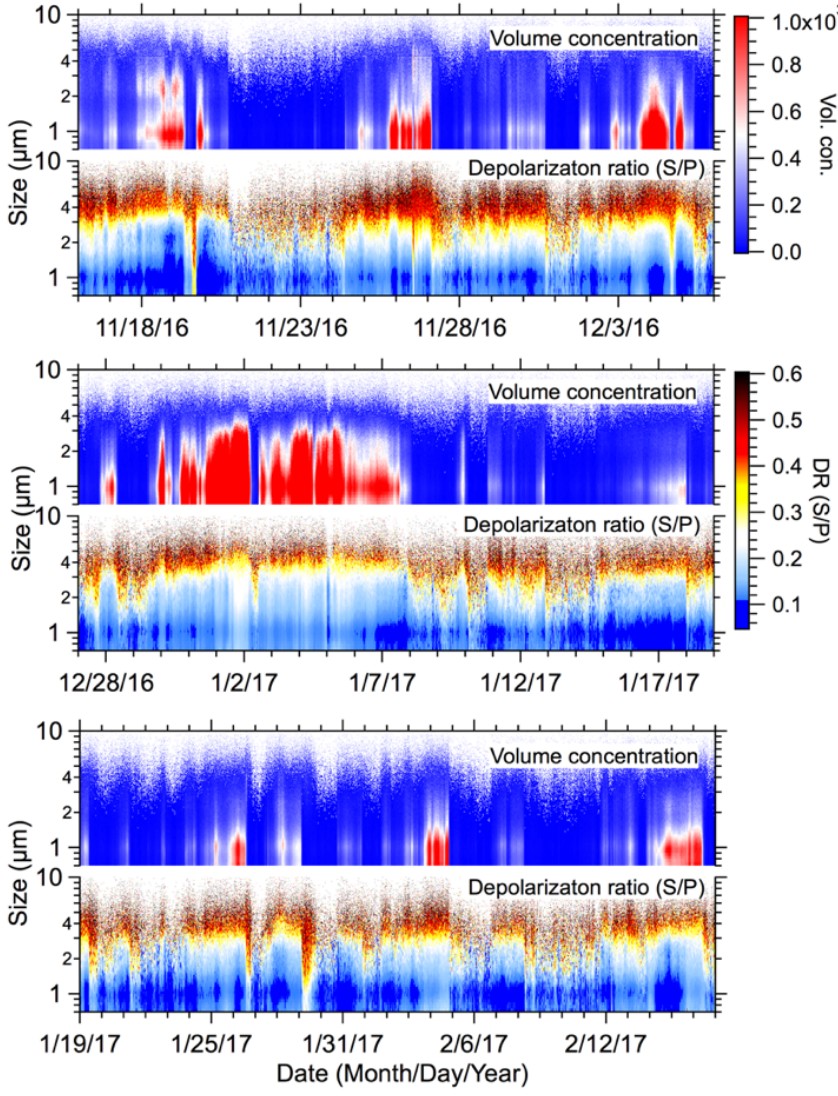

197    Figure 4 Volume concentrations of particles and corresponding depolarization ratio as a function of time during observation

198    period. For better view the performance of the instrument, the observation results are shown in three-time slots.

199











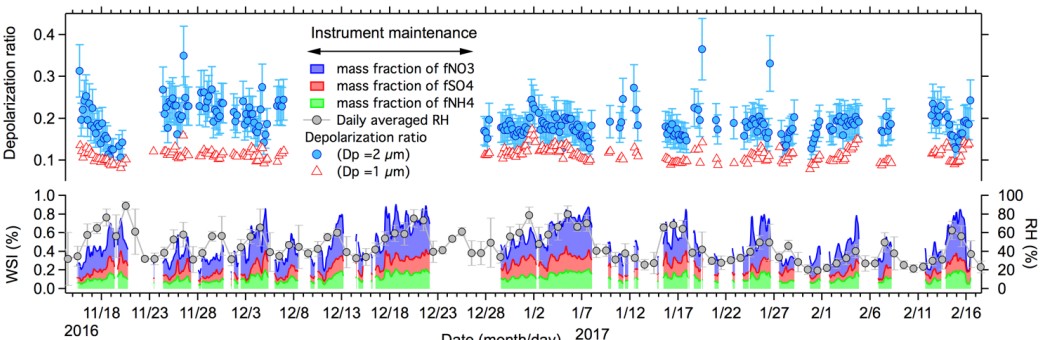


Figure 5 Temporal variabilities of δ value for particles at size of 1 μm and 2 μm (a) and mass fraction of water-soluble
inorganic matter (WSI, Blue: nitrate; Red: sulphate; Green: ammonium) in PM$_{2.5}$, ambient RH (b).















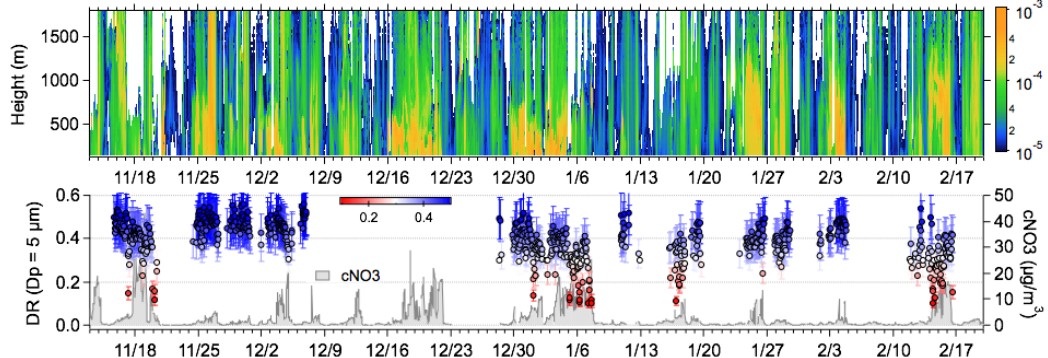


Figure 6 Vertical profile of extinction coefficient of dust particles derived from Lidar observation (a) and variability of $\delta$
value of mineral dust particles and $NO_3^-$ in coarse mode aerosols (b) as a function of time. Here we choose the particles at a
size of 5 μm to indicate the mineral dust particles, marked as $\delta_{Dp=5}$ correspondingly.





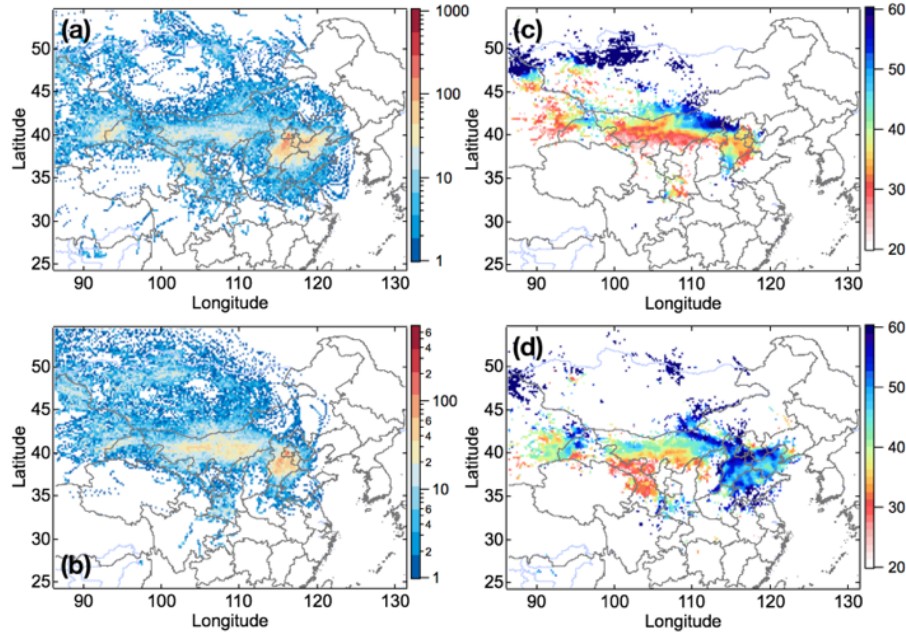

Figure 7 Footprint regions of mineral dust particles for the periods when averaged mass concentration of cNO3 was higher
than 5 μg/m$^3$, while $\delta_{Dp=5}$ values were higher than 0.4 (a) and lower than 0.2 (b), and corresponding mean RH (c) and (d) in
each grid within the footprint area on the basis of ensemble Hysplit analysis. Footprint height was defined as a height that
endpoint of backward trajectory was lower than the height of mixing layer in that grid.






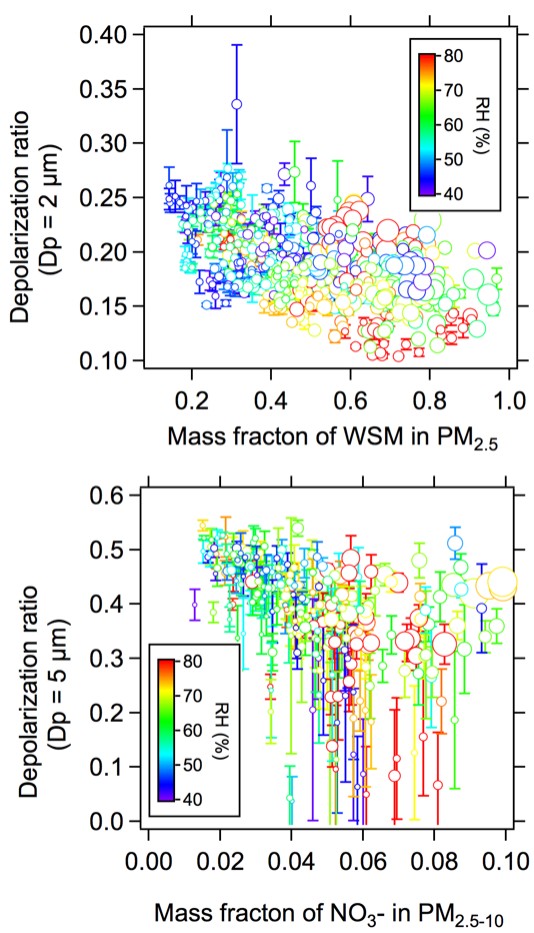


Figure 8 The variation of $\delta_{Dp=2}$ as a function of mass fraction of WSM in PM$_{2.5}$ (a), and $\delta_{Dp=5}$ as a function of mass fraction of cNO3 in PM$_{2.5-10}$. The color represents the hourly averaged relative humidity during measurement.

