# Peer review of "Synergistic effect of water-soluble species and relative humidity on"

_Atmospheric Chemistry and Physics, 2018_

## Referee Comment (RC1) · Anonymous Referee #2 · 17 Sep 2018

Comment on "Synergistic effect of water-soluble species and relative humidity on morphological changes of aerosol particles in Beijing mega-city during severe pollution episodes" by Xiaole Pan et al.

More and more attentions have been paid on air pollution (haze) due to their significant impacts on environment and climate. The article investigates the transformation of non-sphericity of mineral dust particles mixed with pollutants in Beijing based on combined observation from ACSA-14 and POPC. The topic is of sufficient interest to the communities of study of atmospheric aerosol, especially dust and air pollutants. In

general, I find this manuscript to be of interest for publication and appropriate for this journal. There are several suggestions for improvement listed below that should be considered by the authors before publication.

1. Until now there are lots of literatures introduced change of dust morphology during long-range transportation due to mixture with other substances over the past decades, not only based on on-line measurements (e.g., lidar remote sensing), but also in the laboratory analysis (e.g., SEM/TEM). In this paper, the authors should more clearly point out the key point of current study. It will be easier for readers to understand if the authors briefly summary some related studies in section Introduction.

2. Previous studies generally use volume depolarization ratio from lidar measurements to distinguish dust aerosols from others. Of course, POPC have its remarkable advantages. However, it should be noted that there is uncertainty when size of individual particles is estimated from POPC based on forward scattering signal at 60-degree, especially for non-spherical particles (like dust aerosols). So, the authors should discuss and/or introduce the uncertainty of particle size. And it is suggested that the authors should select two large interval sizes when discuss difference of DR at different size.

3. DR not only depend on shape of the particle, but also its size. In this study, authors discuss the difference of DR at different size, to prove change of dust morphology when they mix with other aerosols under high moisture condition. As shown in Fig. 4, it seems there is an obvious peak of DR between 3-5 $\mu$m during the period of field campaign. So, in my opinion it should be careful to conclude.

4. Page 1 line 20: please delete "of" from "February of 2017".

5. Page 4 line 99: change "Februarys" to "February".

6. Page 6 line 87: change "in January29, 2017" to "on January 29, 2017".

7. Page 17 line 85: add "(b)" at the end of the sentence.

8. Aspect ratio is a key parameter for evaluating radiative effects of particles. The authors are encouraged to estimate this parameter from POPC observation in the future.

9. Page 6 line 87: change "in January29, 2017" to "on January 29, 2017".

10. Figure 6: it seems that backscattering coefficient is show in the upper panel according to the order of magnitude in colorbar. Please check carefully what is it. Besides, it is better to add volume depolarization ratio of aerosols from lidar measurements in the figure, so that the readers will be easier to understand the results.

11. A paper about the effects of sulfuric acid and ammonium sulfate coatings on dust aerosols (Eastwood M. et al., 2009) was published in Geophys. Res. Lett.. Please reference this paper to increase reader understanding of interaction between dust and other aerosols. Furthermore, real-time transformation of dust aerosols morphology studied based on ground-based polarization-Raman lidar measurements (Huang Z. et al., 2018, Remote Sensing), will be very useful for readers to understand importance of depolarization ratio for aerosols investigation.

Please also note the supplement to this comment:
https://www.atmos-chem-phys-discuss.net/acp-2018-623/acp-2018-623-RC1-supplement.pdf

---

## Referee Comment (RC2) · Anonymous Referee #1 · 29 Sep 2018

Pan et al., investigated the chemical composition and depolarization ratios of aerosol particles at an urban site in Beijing between November 2016 and February 2017. One important finding is that the depolarization ratio decreased for coarse particles when the mass fraction of nitrate increased. The work presented in this paper can be important for understanding the evolution of physicochemical properties of mineral dust. Nevertheless, the discussion should be further enhanced; in addition, the language should be improved. I can recommend it for publication after my comments have been addressed.

**Major comments:**

Major comment #1, Line 143-145, page 5: Since the aerosol flow was further diluted by zero air (920 ccm, 38% RH), what was the RH of the flow after dilution? I assume that the POPC measured the depolarization ratios at ~38% RH since the dilution factor by the zero air was >10. This information is critical because the morphology and measured depolarization ratios of aerosol particles depend on RH. Please clarify it.

Major comment #2: Section 3.5: I consider this section as the most important part of this manuscript. However, the discussion is far too short and rather descriptive. A number of previous studies have investigated the hygroscopic properties of $Ca(NO_3)_2$ aerosol particles, and I would refer the authors to these papers (Gibson et al., 2006; Guo et al., 2018). I can understand depolarization ratio depended on nitrate fraction (as shown in Figure 8), but what was it also affected by RH? It is related to dependence of aerosol liquid water content on RH? This question is also related to my first major comment: did the author measured the depolarization ratio at ambient RH or at the RH after the aerosol flow was diluted by zero air?

**Minor comments:**

Line 24-25, page 1: When depolarization ratio decreases, does the particle become more spherical or more non-spherical? In the abstract a short introduction to this parameter should be included, though more information of this parameter can be found in page 3.

Line 53-54, page 2: A number of seminal papers (Krueger et al., 2003; Laskin et al., 2005; Tang et al., 2016) on $HNO_3$-$CaCO_3$ reactions should be cited here.

Line 55, page 2: For dust-cloud interactions, two studies (Sullivan et al., 2010; Tang et al., 2015) measured the CCN activity of $CaCO_3$ and its aging products and should be cited here.

Line 361, page 12: please include proper literature.

Line 363, page 12: To support their claim, the author should cite previous studies (Sullivan et al., 2009; Ma et al., 2013; Gu et al., 2018) which showed the hygroscopicity of CaSO4 is very low.

Line 347-349, page 11-12: A recent study (Wu et al., 2018) which explored the aerosol liquid water content and its impact on secondary particle formation should be cited and discussed here.

**Technical comments:**

The language should be further improved, and here I only list a few (but there are more):

Line 42, page 2: change "emit" to "emitted" or "have emitted".

Line 99, page 4: change "was consecutively suffered from" to "consecutively suffered from".

Line 152, page 5: change "note" to "noted".

Line 175, page 6: please change "in consistent well" to "in good consistence".

Line 210, page 6: please change "Number of" to "A number of".

Line 211, page 6: change "was" to "were".

**References:**

Gibson, E. R., Hudson, P. K., and Grassian, V. H.: Physicochemical properties of nitrate aerosols: Implications for the atmosphere, J. Phys. Chem. A, 110, 11785-11799, 2006.

Gu, W. J., Li, Y. J., Zhu, J. X., Jia, X. H., Lin, Q. H., Zhang, G. H., Ding, X., Song, W., Bi, X. H., Wang, X. M., and Tang, M. J.: Investigation of water adsorption and hygroscopicity of atmospherically relevant particles using a commercial vapor sorption analyzer, Atmos. Meas. Tech., 10, 3821-3832, 2017.

Guo, L. Y., Gu, W. J., Peng, C., Wang, W. G., Li, Y. J., Zong, T. M., Tang, Y. J., Wu, Z. J., Lin, Q. H., Ge, M. F., Zhang, G. H., Hu, M., Bi, X. H., Wang, X. M., and Tang, M. J.: A comprehensive study of hygroscopic properties of calcium- and magnesium-containing salts: implication for hygroscopicity of mineral dust and sea salt aerosols, Atmos. Chem. Phys. Discuss., 2018, 1-37, 10.5194/acp-2018-412, 2018.

Krueger, B. J., Grassian, V. H., Laskin, A., and Cowin, J. P.: The Transformation of Solid Atmospheric Particles into Liquid Droplets through Heterogeneous Chemistry: Laboratory Insights into the Processing of Calcium Containing Mineral Dust Aerosol in the Troposphere,

Geophys. Res. Lett., 30, 1148, doi: 1110.1029/2002gl016563, 2003.

Laskin, A., Iedema, M. J., Ichkovich, A., Graber, E. R., Taraniuk, I., and Rudich, Y.: Direct Observation of Completely Processed Calcium Carbonate Dust Particles, Faraday Discuss., 130, 453-468, 2005.

Ma, Q., He, H., Liu, Y., Liu, C., and Grassian, V. H.: Heterogeneous and multiphase formation pathways of gypsum in the atmosphere, Phys. Chem. Chem. Phys., 15, 19196-19204, 2013.

Sullivan, R. C., Moore, M. J. K., Petters, M. D., Kreidenweis, S. M., Roberts, G. C., and Prather, K. A.: Effect of Chemical Mixing State on the Hygroscopicity and Cloud Nucleation Properties of Calcium Mineral Dust Particles, Atmos. Chem. Phys., 9, 3303-3316, 2009.

Tang, M. J., Whitehead, J., Davidson, N. M., Pope, F. D., Alfarra, M. R., McFiggans, G., and Kalberer, M.: Cloud Condensation Nucleation Activities of Calcium Carbonate and its Atmospheric Ageing Products, Phys. Chem. Chem. Phys., 17, 32194-32203, 2015.

Tang, M. J., Cziczo, D. J., and Grassian, V. H.: Interactions of Water with Mineral Dust Aerosol: Water Adsorption, Hygroscopicity, Cloud Condensation and Ice Nucleation, Chem. Rev., 116, 4205–4259, 2016.

Wu, Z. J., Wang, Y., Tan, T. Y., Zhu, Y. S., Li, M. R., Shang, D. J., Wang, H. C., Lu, K. D., Guo, S., Zeng, L. M., and Zhang, Y. H.: Aerosol Liquid Water Driven by Anthropogenic Inorganic Salts: Implying Its Key Role in Haze Formation over the North China Plain, Environ. Sci. Tech. Lett., 5, 160-166, 2018.

---

## Author Comment (AC1) · 8 Nov 2018

The authors appreciate the anonymous reviewer very much for providing insight comments on the manuscript, we would like to revise the context on the basis of suggestions and advises carefully. The specific replies are as follows:

Major comment #1, Line 143-145, page 5: Since the aerosol flow was further diluted by zero air (920 ccm, 38% RH), what was the RH of the flow after dilution? I assume that the POPC measured the depolarization ratios at ∼38% RH since the dilution fac-

[Figure]

tor by the zero air was >10. This information is critical because the morphology and measured depolarization ratios of aerosol particles depend on RH. Please clarify it.

Reply: we agree with the comment that physical property of aerosol is critically depend on the humid condition, and mixing of sampling aerosol flow with dry diluting flow will change the morphology of particles. For the measurement of instruments, the exhausted air from measuring chamber was filtered and recycled to sever as diluting and sheath flow. Since the sampling flow rate was much lower than the sum of diluting and sheath flow air, relative humidity of the flow after dilution was the same as the dilution flow. As the reviewer mentioned, the original morphology of particle in the measuring chamber might be altered as a result of mixing with dry air. Besides, the temperature in the measuring chamber was ~4°C higher than the ambient temperature which also lead to decrease of relative humidity. Therefore, the depolarization ratio in this study refer to the value in the dry condition. We will clarify this point in the revised manuscript.

Major comment #2: Section 3.5: I consider this section as the most important part of this manuscript. However, the discussion is far too short and rather descriptive. A number of previous studies have investigated the hygroscopic properties of Ca(NO3)2 aerosol particles, and I would refer the authors to these papers (Gibson et al., 2006; Guo et al., 2018). I can understand depolarization ratio depended on nitrate fraction (as shown in Figure 8), but what was it also affected by RH? It is related to dependence of aerosol liquid water content on RH? This question is also related to my first major comment: did the author measured the depolarization ratio at ambient RH or at the RH after the aerosol flow was diluted by zero air?

Reply: As suggested by reviewer, the relationship between depolarization ratio of particles in coarse mode and the relative humidity is an important indicator for investigating the heterogeneous reaction on the surface of dust particles. During the observation period, the relative humidity in the measuring chamber was stable at 34.3 ± 1.6 %, indicating that physical properties of particles in measuring chamber would have been altered, in particular at high ambient RH condition (Before 2017 January 9). Therefore, the depolarization ratio of the particles reported in this study was representative of that at relatively dry condition, not the ambient condition. Nevertheless, some water-soluble species could absorb water at very low relative humidity condition (Gibson et al., 2006). Take Ca(NO3)2 for instance, it would be deliquescent at relative humidity less than 10%, which means that they may still exist in liquid phase at relative humidity of ∼34%, though efflorescent process may reduce its sphericity. In Beijing, the major water soluble composition of the particles in coarse mode was nitrate, as reported by numbers of previous studies. Presuming the counterpart of nitrate was calcium, the water content could be calculated, and the influence of reduction in relative humidity in the measuring chamber could be also evaluated.

As shown in Fig.1, the mass concentration of H2O as a result of deliquescent effect of nitrate in coarse mode was calculated. During the high cNO3 and relative humidity condition, the water content of particle in coarse mode reduced by ∼40% on average, accounting for less 10% of total mass of particle in coarse mode. The percentage reduction in diameter of particles was calculated to be less than 5%, implying that the evaporation of water in the measuring chamber did not change our main conclusion. As suggested, now we are planning an experiment using HTAAC (humidifying Tandem Aerodynamic Aerosol Classifier) in laboratory to check the relationship between depolarization ratio of dust particles and relative humidity, and the modification of morphology of dust particle due to reaction with nitric acid will be also considered.

Minor comments:

Line 24-25, page 1: When depolarization ratio decreases, does the particle become more spherical or more non-spherical? In the abstract a short introduction to this parameter should be included, though more information of this parameter can be found in page 3.

Reply: The information will be included in introduction section.

Line 53-54, page 2: A number of seminal papers (Krueger et al., 2003; Laskin et al.,

2005; Tang et al., 2016) on HNO3-CaCO3 reactions should be cited here.

Reply: The literatures will be cited in revised manuscript.

Line 55, page 2: For dust-cloud interactions, two studies (Sullivan et al., 2010; Tang et al., 2015) measured the CCN activity of CaCO3 and its aging products and should be cited here.

Reply: The literatures will be cited in revised manuscript.

Line 361, page 12: please include proper literature.

Reply: Relevant literatures will be cited in revised manuscript.

Line 363, page 12: To support their claim, the author should cite previous studies (Sullivan et al., 2009; Ma et al., 2013; Gu et al., 2018) which showed the hygroscopicity of CaSO4 is very low.

Reply: Relevant literatures will be cited in revised manuscript.

Line 347-349, page 11-12: A recent study (Wu et al., 2018) which explored the aerosol liquid water content and its impact on secondary particle formation should be cited and discussed here.

Reply: Relevant literatures will be cited in revised manuscript.

Technical comments: Line 42, page 2: change "emit" to "emitted" or "have emitted".
 Line 99, page 4: change "was consecutively suffered from" to "consecutively suffered from".
 Line 152, page 5: change "note" to "noted".
 Line 175, page 6: please change "in consistent well" to "in good consistence". Line 210, page 6: please change "Number of" to "A number of".
 Line 211, page 6: change "was" to "were".

Reply: All the Technical errors will be corrected in the revised manuscript, and we will check all grammar and expression in English throughout the content.

**Fig. 1.** Temporal variations in mass concentration of nitrate in coarse mode (black line), relative humidity (gray line), calculated water content in coarse mode at ambient condition (red line with circ

---

## Author Comment (AC2) · 8 Nov 2018

The authors appreciate the anonymous reviewer very much for providing insight comments on the manuscript, we would like to revise the context on the basis of suggestions and advises carefully. The specific replies are as follows:

1. Until now there are lots of literatures introduced change of dust morphology during long-range transportation due to mixture with other substances over the past decades, not only based on on-line measurements (e.g., lidar remote sensing), but also in the

[Figure]

laboratory analysis (e.g., SEM/TEM). In this paper, the authors should more clearly point out the key point of current study. It will be easier for readers to understand if the authors briefly summary some related studies in section Introduction.

Reply: We will briefly summarize the advances in morphological studies of particles using LIDAR and electro-microscopy in introduction section.

2. Previous studies generally use volume depolarization ratio from lidar measurements to distinguish dust aerosols from others. Of course, POPC have its remarkable advantages. However, it should be noted that there is uncertainty when size of individual particles is estimated from POPC based on forward scattering signal at 60-degree, especially for non-spherical particles (like dust aerosols). So, the authors should discuss and/or introduce the uncertainty of particle size. And it is suggested that the authors should select two large interval sizes when discuss difference of DR at different size.

Reply: We agree with the comments of reviewer, and the uncertainty of measurement in size of particle will be introduce in the revised manuscript. As suggested, ddifference of DR for two large interval size will be discussed.

3. DR not only depend on shape of the particle, but also its size. In this study, authors discuss the difference of DR at different size, to prove change of dust morphology when they mix with other aerosols under high moisture condition. As shown in Fig. 4, it seems there is an obvious peak of DR between 3-5 $\mu$m during the period of field campaign. So, in my opinion it should be careful to conclude.

Reply: As suggested, the DR value of particles are critically depend on their size. Larger particles have larger DR value, consistent with the model simulation. We will carefully conclude this point in the revised manuscript.

4. Page 1 line 20: please delete "of" from "February of 2017".

Reply: The word "of" in "February of 2017" will be deleted in the revised manuscript.

5. Page 4 line 99: change "Februarys" to "February".

Reply: The misspelling will be corrected in the revised manuscript.

6. Page 6 line 87: change "in January29, 2017" to "on January 29, 2017".

Reply: "in" will be changed to "on".

7. Page 17 line 85: add "(b)" at the end of the sentence.

Reply: "(b)" will be added.

8. Aspect ratio is a key parameter for evaluating radiative effects of particles. The authors are encouraged to estimate this parameter from POPC observation in the future.

Reply: We will estimate the aspect ratio of the particle according to T-matrix simulations.

9. Page 6 line 87: change "in January29, 2017" to "on January 29, 2017".

Reply: "in" will be changed to "on".

10. Figure 6: it seems that backscattering coefficient is show in the upper panel according to the order of magnitude in colorbar. Please check carefully what is it. Besides, it is better to add volume depolarization ratio of aerosols from lidar measurements in the figure, so that the readers will be easier to understand the results.

Reply: We confirmed that the upper panel shows the vertical profile of extinction coefficient determined by Lidar. We will add depolarization ratio information in the revised manuscript.

11. A paper about the effects of sulfuric acid and ammonium sulfate coatings on dust aerosols (Eastwood M. et al., 2009) was published in Geophys. Res. Lett.. Please reference this paper to increase reader understanding of interaction between dust and other aerosols. Furthermore, real-time transformation of dust aerosols morphology studied based on ground-based polarization-Raman lidar measurements (Huang Z. et al., 2018, Remote Sensing), will be very useful for readers to understand importance

of depolarization ratio for aerosols investigation.

Reply: The relevant literatures will be cited in the revised manuscript.